Xenoposeidon is the earliest known rebbachisaurid sauropod dinosaur

http://orcid.org/0000-0002-1003-5675 Taylor Michael P. dino@miketaylor.org.uk
Department of Earth Sciences, University of Bristol , Bristol , England
Young Mark
Electronic publication date: 2018 Jul 6
Publication date: 2018
Volume: 6
Electronic Location ID: e5212
Received 2017 Nov 15; Accepted 2018 Jun 20
Copyright: © 2018 Taylor
Copyright year: 2018
Copyright holder: Taylor
License: This is an open access article distributed under the terms of the Creative Commons Attribution License, which permits unrestricted use, distribution, reproduction and adaptation in any medium and for any purpose provided that it is properly attributed. For attribution, the original author(s), title, publication source (PeerJ) and either DOI or URL of the article must be cited.
License URL: https://creativecommons.org/licenses/by/4.0/

Keywords: Xenoposeidon, Rebbachisauridae, Sauropoda, Dinosauria, Laminae, Wealden, 3D modelling

Funding: The authors received no funding for this work.

==============================
Xenoposeidon proneneukos is a sauropod dinosaur from the Early Cretaceous Hastings Group of England. It is represented by a single partial dorsal vertebra, NHMUK PV R2095, which consists of the centrum and the base of a tall neural arch. Despite its fragmentary nature, it is recognisably distinct from all other sauropods, and is here diagnosed with five unique characters. One character previously considered unique is here recognised as shared with the rebbachisaurid diplodocoid Rebbachisaurus garasbae from the mid-Cretaceous of Morocco: an ‘M’-shaped arrangement of laminae on the lateral face of the neural arch. Following the more completely preserved R. garasbae, these laminae are now interpreted as ACPL and lateral CPRL, which intersect anteriorly; and PCDL and CPOL, which intersect posteriorly. Similar arrangements are also seen in some other rebbachisaurid specimens (though not all, possibly due to serial variation), but never in non-rebbachisaurid sauropods. Xenoposeidon is therefore referred to Rebbachisauridae. Due to its inferred elevated parapophysis, the holotype vertebra is considered a mid-posterior dorsal despite its elongate centrum. Since Xenoposeidon is from the Berriasian–Valanginian (earliest Cretaceous) Ashdown Formation of the Wealden Supergroup of southern England, it is the earliest known rebbachisaurid by some 10 million years. Electronic 3D models were invaluable in determining Xenoposeidon’s true affinities: descriptions of complex bones such as sauropod vertebrae should always provide them where possible.

Introduction

The fossil record of sauropod dinosaurs extends through most of the Mesozoic, from the Late Triassic (Lallensack et al., 2017) to the very end of the Cretaceous (Riera et al., 2009; Sellés et al., 2016). However, their record in the earliest Cretaceous, as for most dinosaurs, is much less rich (Tennant et al., 2017). In fact, almost the entire record of sauropodomorphs in the first three ages of the Cretaceous rests on fossils from Europe (Tennant, Chiarenza & Baron, 2018: fig. 11: parts C–F)—only Leinkupal from the Berriasian/Valanginian of Argentina (Gallina et al., 2014), and perhaps Euhelopus (based on a revised age for the Mengyin Formation, Borinder, Poropat & Kear, 2016) have been named in this period from outside of Europe. In this context, sauropods from earliest Cretaceous formations in Europe are particularly important for our understanding of the evolution of this group.

Xenoposeidon proneneukos is a neosauropod sauropod dinosaur from the Berriasian–Valanginian (earliest Cretaceous) Ashdown Formation of the Wealden Supergroup of southern England (Taylor & Naish, 2007). It is represented by a single partial mid-to-posterior dorsal vertebra, NHMUK PV R2095 (Fig. 1; BMNH R2095 at the time of the original description by Taylor & Naish, 2007; NHMUK R2095 at the time of Taylor, 2017). This element consists of the centrum and the base of a tall neural arch, broken off below the transverse processes and zygapophyses. Despite its fragmentary nature, it is recognisably different from all other sauropods, and Taylor & Naish (2007) diagnosed it on the basis of six characters that they considered unique among sauropods. Here, I will present a revised diagnosis.

Figure 1 NHMUK R2095, the holotype and only vertebra of Xenoposeidon proneneukos, shown from all six cardinal directions.

(A) Dorsal view, with anterior to the left. (B) anterior, (C) left lateral, (D) posterior, (E) right lateral view and (F) ventral view, with anterior to the left. Scale bar = 200 mm.

Taylor & Naish (2007: 1554–1557) compared the Xenoposeidon vertebra to those of the main neosauropod groups—Diplodocoidea, Camarasauridae, Brachiosauridae and Titanosauria—and concluded that it could not be convincingly referred to any of these groups (see Fig. 2). Their phylogenetic analysis (pp. 1157–1558 and fig. 6) corroborated this by recovering Xenoposeidon as a neosauropod in all most parsimonious trees, but in a polytomy with all other neosauropods, wholly unresolved save that the clade Flagellicaudata was preserved in all MPTs.

Figure 2 Comparative morphology of mid-posterior dorsals from six sauropods: Xenoposeidon and five representatives of major groups.

Each vertebra is shown in anterior and left lateral (or right lateral reversed) views, scaled to the same centrum height. (A) The diplodocid Diplodocus carnegii CM 84, 8th dorsal vertebra: (A1) anterior, modified from Hatcher (1901: plate VIII), (A2) right lateral reversed, modified from Hatcher (1901: plate VII). (B) The rebbachisaurid Rebbachisaurus garasbae MNHN MRS 1958, posterior dorsal vertebra: (B1) anterior, (B2) left lateral. (C) Xenoposeidon proneneukos NHMUK R2095, mid-posterior dorsal vertebra: (C1) anterior, (C2) left lateral. (D) The camarasaurid Camarasaurus supremus AMNH 5760/D-X-125, ?10th dorsal vertebra, modified from Osborn and Mook (1921: plate LXX): (D1) anterior, (D2) left lateral. (E) The brachiosaurid Giraffatitan brancai MB.R.3822 (formerly HMN AR1), from a digital model supplied by Heinrich Mallison: (E1) anterior, (E2) right lateral reversed. (F) The titanosaur Yongjinglong datangi GSGM ZH(08)-04, mid-dorsal vertebra, modified from Li et al. (2014: figure 9): (F1) anterior, (F2) left lateral.

In light of Wilson & Allain’s (2015) redescription of the rebbachisaurid diplodocoid Rebbachisaurus garasbae from the mid-Cretaceous of Morocco, and the availability of more photographs and models of rebbachisaurid material, it has now become possible to reinterpret the idiosyncratic system of laminae found in Xenoposeidon, and to refer it confidently to an existing family-level clade.

Validity of Xenoposeidon

Upchurch, Mannion & Barrett (2011: 497–498), in a review of Wealden sauropods, reassessed Xenoposeidon, accepting its validity and concurring with Taylor & Naish (2007) that it was difficult to place within any recognised sauropod clade. However, they tentatively proposed a basal somphospondylan identity for it. Similarly, Mannion et al. (2013: 151) tentatively considered it most likely a basal macronarian.

D’Emic (2012: 651) asserted that ‘the absence of diagnostic features renders Xenoposeidon a nomen dubium’. However, his assessment was mistaken in several respects. For example, the extension of the base of the neural arch to the posterior extremity of the centrum is clearly not, as he asserted, due to damage. D’Emic claimed that dorsal vertebrae illustrated by Osborn & Mook (1921: plates LXIX and LXXII) have forward-sloping neural arches resembling those of Xenoposeidon: in reality, only one posterior dorsal vertebrae out of four complete dorsal columns illustrated in that monograph shows a forward slope, and it differs so much from its fellows that this can only be interpreted as the result of crushing. D’Emic further claimed that the lamina patterns observed in Xenoposeidon can be recognised in other sauropods, but I have been unable to find morphology resembling them in the descriptions he suggests: Osborn and Mook 1921 for Camarasaurus, Riggs, 1903 for Brachiosaurus (probably a typo for Riggs, 1904, which also does not depict similar patterns), Carballido et al. (2011) for Tehuelchesaurus. However, a similar pattern does appear in Rebbachisaurus, as will be discussed below. D’Emic (2012: 651) is probably correct that the ‘asymmetric neural canal’ described by Taylor & Naish (2007: 1553–1554) is a misreading of the tall anterior fossa as being the anterior portion of the neural canal: as Taylor and Naish pointed out, ‘The vacuity is filled with matrix, so the extent of its penetration posteriorly into the neural arch cannot be assessed’. Nevertheless, the shape and size of the fossa is unique among sauropods, and it is bounded by laminae which do not seem to be medial CPRLs—see below. In summary, as will be shown in more detail below, X. proneneukos is a valid, diagnosable taxon, contra D’Emic (2012).

Revised diagnosis

Xenoposeidon differs from all other known sauropods in five respects. Compare the following characters with the state in mid-posterior dorsal vertebra of other sauropods as shown in Fig. 2.

Neural arch covers dorsal surface of centrum. The posterior margin is continuous with that of the centrum, such that in lateral view the posterior margin of the vertebra forms a single smooth curve (Fig. 3: 1a, 1b). In most sauropod dorsals, the base of the neural arch is some way forward of the posterior margin of the centrum. Even in R. garasbae, where the posterior margin of the neural arch approaches that of the centrum, there is a distinct kink in lateral view between the posteroventral slope of the ventral part of the arch’s posterior border, and the vertical margin of the centrum (Fig. 4B).

Neural arch slopes anteriorly 30°–35° relative to the vertical, as determined by the orientation of the posterior articular surface of the centrum (Fig. 3: 2). In fully lateral view, vertical orientation of the posterior articular surface is difficult to determine because the bone extends slightly further posteriorly at centrum mid-height than more dorsally or ventrally, but it is easy to see in a slightly posterolateral view, as can determined from the 3D model (File S1).

Sharp oblique lamina above lateral fossa forms ventral border of a broad, flat area of featureless bone. The fossa beneath this ridge-like lamina (Fig. 3: 3a) contains nested within it a deeper lateral foramen; and above it, below the ‘M’-shaped complex of laminae that are discussed in detail below, the bone is quite flat and smooth (Fig. 3: 3b).

Very large, teardrop-shaped anterior fossa, nearly as tall as the posterior articular facet of the centrum and half as transversely broad as it is tall (Fig. 3: 4). In R. garasbae, the fossa is proportionally nearly as tall, but much narrower (Wilson & Allain, 2015: fig. 3E).

Arched laminae form vaulted boundary of anterior fossa. These laminae (Fig. 3: 5a) cannot be interpreted as the medial CPRLs (Fig. 3: 5b), as those arise separately from the neural arch pedicels. These laminae arising from the pedicels cannot instead be regarded lateral CPRLs, as those laminae are located on the lateral face of the neural arch, as will be discussed below. Furthermore, the point where the arched supporting laminae meet at the top of their arch is located some way ventral to the location inferred for the prezygapophyses based on the trajectory of the preserved portions of the medial CPRLs (Fig. 5).

Figure 3 Autapomorphies of Xenoposeidon proneneukos NHMUK R2095, mid-posterior dorsal vertebra, highlighted in red.

(A) Anterior view. (B) Left lateral view. Numbers pertain to the numbering of autapomorphies in the text. (1a), neural arch covers whole of centrum, and (1b) is contiguous with posterior articular facet. (2), neural arch is inclined forward by 35° relative to the vertical. (3a), inclined ridge-like lamina marks ventral margin of (3b) broad featureless area of bone. (4), large teardrop-shaped anterior fossa. (5a), vaulted laminae bound this fossa, but are not the medial CPRLs (5b, drawn in finer lines), which continue up to the presumed location of the prezygapophyses.

Figure 4 Centra and neural arches of posterior dorsal vertebrae from two rebbachisaurid sauropods (not to scale), highlighting the distinctive ‘M’ shape formed by laminae on the lateral face of the neural arch.

(A) NHMUK R2095, the holotype and only vertebra of Xenoposeidon proneneukos. (B) MNHN MRS 1958, a posterior dorsal vertebra from the holotype specimen of Rebbachisaurus garasbae.

Figure 5 NHMUK R2095, the holotype and only vertebra of Xenoposeidon proneneukos, in left anteroventrolateral view, highlighting the three sets of laminae related to the prezygapophyses.

The trajectories of the medial CPRLs (which emerge from the neural arch pedicels) and the lateral CPRLs (which intersect with the APCLs) indicate the approximate position of the prezygapophyses. The additional arched laminae form the margins of the large, teardrop-shaped anterior fossa, homologous with a CPRF, but meet at a position some way below and posterior to the presumed location of the prezygapophyseal facets. Breakage of both medial CPRLs and the left ACPL and PCDL is indicated by cross-hatching. Note that, from this perspective, the lateral CPRL appears to turn a corner where it intersects with the ACPL, such that the posteroventral portion of the lateral CPRL appears contiguous with the dorsal portion of the ACPL. This is an illusion brought about by the eminence at the point of intersection. As always, this is much easier to see in three dimensions (see File S1).

Reinterpretation of Xenoposeidon

Taylor & Naish’s (2007) history, geography, geology and description of the Xenoposeidon specimen require no revision, and should continue to be considered definitive: this paper does not supersede the earlier description, but should be read in conjunction with it.

The illustrations of the specimen in the original paper, however, were in monochrome and omitted the dorsal and ventral views. The present paper supplements these illustrations with a colour depiction from all six cardinal directions (Fig. 1), an oblique view (Fig. 5) and a high-resolution 3D model of the specimen (File S1).

More importantly, Taylor & Naish’s (2007) interpretation of some features of the vertebra, particularly the ‘M’-shaped complex of laminae on the lateral faces of the neural arch, was mistaken. Although the neural spine and dorsal part of the neural arch are missing, including the pre- and post- zygapophyses and lateral processes, they wrote that ‘sufficient laminae remain to allow the positions of the processes to be inferred with some certainty’. But their inferences were incorrect. Taylor & Naish (2007: 1553) interpreted the cross-shaped structure on the anterodorsal part of the left lateral face of the neural arch as the site of the parapophysis, despite the lack of any articular facet in that location. This influenced their interpretation of the four laminae that met at that point as the ACPL below, the PPDL above, the PRPL anteriorly and an unnamed accessory infraparapophyseal lamina posteroventrally, which they interpreted as homologous with a PCPL (Fig. 6A). Similarly, they did not attempt to identify either the long lamina running up the posterior edge of the lateral face of the neural arch (designating it only ‘posterior lamina’) or the lamina forming a shallow ‘V’ with the ‘accessory infraparapophyseal lamina’, simply calling it an ‘accessory postzygapophyseal lamina’ (Fig. 6A).

Figure 6 NHMUK R2095, the holotype and only vertebra of Xenoposeidon proneneukos, in left lateral view, with interpretative drawings.

(A) The incorrect interpretation of the laminae from Taylor & Naish (2007: figure 4A), with identifying captions greyed out since they are largely incorrect. (B) The revised interpretation of the same laminae, based on the similar arrangement in Rebbachisaurus garasbae. Scale bar = 200 mm.

Among the various unusual features of the Xenoposeidon vertebra, the ‘M’-shaped set of laminae is immediately apparent in lateral view (Fig. 4A): a line can be traced from the anterior margin of the neural arch’s lateral face up the ACPL to the cross that was interpreted as the parapophysis, then posteroventrally down the ‘accessory infraparapophyseal lamina’, then posterodorsally up the ‘accessory postzygapophyseal lamina’ and finally down the posterior margin of the neural arch’s lateral face, along the ‘posterior lamina’. Photographs of other specimens that were available to us at this time did not apparently manifest similar features.

But subsequent work on R. garasbae (Wilson, 2012: 100, fig. 9; Wilson & Allain, 2015)—and an associated video of the rotating vertebra (see Acknowledgements)—show that Rebbachisaurus has a similar complex of laminae (Fig. 4B), which are described by Wilson & Allain (2015: 6) as the second of the eight autapomorphies that they listed for the species: ‘infrazygapophyseal laminae (lat. CPRL, CPOL) that intersect and pass through neighbouring costal laminae (ACPL, PCDL) to form an ‘M’ shape’.

Because the illustrated dorsal vertebra of Rebbachisaurus—MNHN MRS 1958—is substantially complete, it is possible to follow the trajectories of the laminae that participate in the ‘M’ to their apophyses, and so determine their true identities (Fig. 4). The two vertically oriented laminae—the outer pillars of the ‘M’—continue up past the top of the ‘M’. The anterior one supports the parapophysis, and the posterior supports the diapophysis. Also, the two laminae that form the valley in the middle of the ‘M’ support the prezygapophysis and postzygapophysis: in both cases, as noted by Wilson and Allain, they intersect the vertical lamina before continuing to meet their respective zygapophyses. The four laminae that make up the ‘M’, from anterior to posterior, are therefore the ACPL, posterior part of the lateral CPRL, anterior part of the CPOL, and PCDL. Of these, the intersection between the ACPL and lateral CPRL is clearly visible in left lateral view of MNHN MRS 1958 (Fig. 4B). The intersection between the CPOL and PCDL is less apparent in this view, though clear in three dimensions. Both laminae continue dorsally beyond this intersection, but their paths are somewhat changed at the point of contact, with the dorsal portion of the PCDL inclining more anteriorly, and the rod-like CPOL apparently passing through the sheet of bone formed by the PCDL to meet the postzygapophysis.

The referred R. garasbae specimen NMC 50844 described and illustrated by Russell (1996: 388–390 and fig. 30) is also broadly consistent with this morphology. It is not possible to be definite about the laminar intersection based only on line drawings of the specimen from the four cardinal directions, but, as illustrated in Russell’s fig. 30c, the lateral CPRL does appear to pass through the ACPL. The CPOL seems in this specimen to originate posterior to the PCDL, not intersecting with it. However, this difference from the holotype dorsal may be serial variation since, as Russell notes, the relatively longer centrum of his specimen indicates a more anterior serial position than for the holotype’s dorsal vertebra; and this interpretation is corroborated by the observation that, based on lamina trajectories, the anteroposterior distance between the parapophysis and diapophysis was less in NMC 50844 than in the holotype.

In light of these Rebbachisaurus specimens, the mysterious laminae of Xenoposeidon are more readily explained. It is now apparent that the cross on the side of the Xenoposeidon vertebra is not the site of the parapophysis, as Taylor & Naish (2007: 1553) proposed, but merely the intersection of two laminae that pass right through each other: the ACPL, running dorsolaterally, and the lateral CPRL, extending anterodorsally to the (missing) prezygapophysis (Fig. 6B). Similarly, the ‘posterior lamina’ is the PCDL, and it intersects with the CPOL, though the intersection is lost in NHMUK PV R2095 (Fig. 6B). Both the parapophysis and diapophysis of the Xenoposeidon vertebrae would likely have been located some distance above the preserved portion, the former anterior to the latter.

It appears from Dalla Vecchia (1999: fig. 47, left part) that in the holotype and only vertebra of Histriasaurus boscarollii, ‘WN-V6’, the CPOL on the right side of the vertebra intersects with the PCDL in the same way as in Rebbachisaurus, though it is not possible to determine whether the lateral CPRL similarly intersects the ACPL. Dorsal vertebrae of other rebbachisaurid sauropods, however, do not appear to feature the distinctive ‘M’ and intersecting laminae of Rebbachisaurus and Xenoposeidon: The 3D model of a dorsal vertebra of Nigersaurus (Sereno et al., 2007) shows that the lateral CPRLs originate anterior to the ACPLs and the CPOLs posterior to the PCDLs, so that there is no intersection. A subtle ‘V’ shape does appear high up on the lateral faces of the neural arch, between the ACPL and the PCDL, but it seems unrelated to the lateral CPRL and CPOL.

Unpublished 3D models of an anterior dorsal neural arch and a more posterior dorsal vertebra of Katepensaurus (L. M. Ibiricu, 2017, personal communication) as illustrated in figs. 3A and 5A of Ibiricu et al. (2017) show that in both vertebrae, the lateral CPRLs originate anterior to the ACPLs, and the CPOLs seem to originate posterior to the PCDLs—though damage to the posterior portion makes the latter uncertain.

The laminae do not appear to intersect in the illustrated dorsal vertebra of Demandasaurus (Torcida Fernández-Baldor et al., 2011: fig. 9).

The sole known vertebra of Nopcsaspondylus seems to have an entirely different pattern of lamination (Mannion, 2010: fig. 5) with no lamina intersections like those of MNHN MRS 1958.

No determination can be made for other rebbachisaurids as they are insufficiently preserved or illustrated (e.g. Limaysaurus, Amazonsaurus, Cathartesaura), or simply lack posterior dorsal vertebral material (e.g. Rayososaurus, Tataouinea, Comahuesaurus, Zapalasaurus).

However, one cannot rule out the possibility that complete and well-preserved posterior dorsal vertebrae of most or all rebbachisaurids have Rebbachisaurus-like intersecting laminae: even in those species for which a well-preserved vertebra lacks them, this could be due to serial variation, with these features only fully developing in the more posterior dorsals.

Xenoposeidon, then, resembles Rebbachisaurus in the possession of a distinctive ‘M’ on the lateral face of the neural arch, in the intersecting lateral CPRL and ACPL, and in the elevation of the parapophysis above the level of the prezygapophysis, as inferred from the trajectories of the lateral CPRL and ACPL—a complex of related features. Although at first glance they do not closely resemble each other, Xenoposeidon and Rebbachisaurus, while geometrically different, are topologically similar.

A superficially similar ‘M’-shaped complex of laminae is also found in dorsal vertebrae of the saltasaurine titanosaur Neuquensaurus (Salgado, Apesteguía & Heredia, 2005: figs. 3 and 4). However, this is not homologous to the situation in Rebbachisaurus and Xenoposeidon, as different laminae are involved: Salgado, Apesteguía & Heredia (2005: 626) identify the inner arms of the ‘M’ as the PCPL and a novel accessory PCDL which they term the APCDL. (This APCDL, together with the ventral portion of the PCDL proper, constitute the ‘ventrally forked infradiapophyseal lamina’ of Salgado, Coria & Calvo, 1997.) It is apparent from the illustrations of Salgado, Apesteguía & Heredia (2005: figs. 3C and especially 4A–4B) that the APCDL of Neuquensaurus is not contiguous with, and cannot be considered a part of, the CPOL.

Regarding the significance of the elevated parapophysis, since no complete or nearly complete rebbachisaurid dorsal column has been described, comparisons with other, better represented sauropods are warranted. In the probable basal diplodocoid Haplocanthosaurus, the dorsal margin of the parapophyseal facet reaches the level of, and is coincident with, the prezygapophyseal facet around dorsal vertebra 7 or 8, but never rises any higher than this in more posterior vertebrae (Hatcher, 1903: plate I). In the more distantly related diplodocid diplodocoids Apatosaurus and Diplodocus, the parapophysis never migrates far enough dorsally to reach a position level with the prezygapophyses, even in the most posterior dorsals (Gilmore, 1936: plate XXV; Hatcher, 1901: plates VII, VIII).

Taylor & Naish (2007: 1554) argued that Xenoposeidon could not at that time be convincingly referred to Rebbachisauridae because Rebbachisaurus differs from NHMUK PV R2095 in five ways: ‘possession of a very prominent PCDL (mistranslated as PCPL in the original), large and laterally diverging prezygapophyses, depressions at the base of the neural arch (Bonaparte, 1999: 173), lateral foramina not set within fossae, and a strongly arched ventral border to the centrum’. Of these features, the first is now recognised as occurring in Xenoposeidon; the second appears to be an outright error, as the prezygapophyses of Rebbachisaurus meet on the midline, and in any case the situation in Xenoposeidon is not known. ‘Depressions at the base of the neural arch’ seems to be a mistranslation of Bonaparte’s original Spanish, ‘profundas depresiones en la base de la espina neural’, which refers not to the neural arch but the neural spine, and since this portion is not preserved in Xenoposeidon, it is not informative for our purposes. The 3D model of the Rebbachisaurus dorsal suggests that its lateral foramina are set in shallow depressions, but these are far less pronounced than those of Xenoposeidon. This leaves the stronger arching of the ventral border of the centrum in Rebbachisaurus, but this difference is not convincing given that the ventral margin of the NHMUK PV R2095 posterior cotyle is incomplete and the anterior end of the centrum is missing: the ventral border was likely rather more arched when the vertebra was complete.

In conclusion, the weight of morphological evidence, including the camerate internal tissue structure of the centrum that is exposed in anterior view (Fig. 1B), supports including Xenoposeidon within Rebbachisauridae. This is compatible with the observation of Taylor & Naish (2007: 1557), in whose phylogenetic analysis ‘various most-parsimonious trees also recover Xenoposeidon in many other positions, including as a … rebbachisaurid’.

Serial position

The serial position of the R. garasbae holotype dorsal vertebra MNHN MRS 1958 is not definitely known. However, it has been uniformly referred to as a posterior dorsal, most likely due to the very elevated position of its parapophyses and Lavocat’s (1954) initial assessment of it as ‘une des dernières dorsales’ (one of the last dorsals)—perhaps made with knowledge of the spatial relation of bones in the quarry.

The position of the X. proneneukos holotype vertebra NHMUK PV R2095 is of course even more difficult to determine in light of the limited nature of the specimen, though its similarity to MNHN MRS 1958 suggests a similar position. Taylor & Naish (2007: 1553) wrote that ‘the high position of the parapophysis on the neural arch of R2095 indicates a mid to posterior placement of the vertebra within the dorsal column, but, because the prezygapophyses must have been dorsal to it, it was probably not among the most posterior vertebrae in the sequence’. With the location of the parapophysis now interpreted as significantly higher than previously thought, and probably well above the prezygapophysis, an even more posterior position is indicated.

A posterior serial position is surprising in light of the anteroposterior length of the Xenoposeidon centrum. Its posterior articular surface measures 160 mm high by 170 mm wide, while the length of even the preserved portion of the centrum is 190 mm, and it must have been at least 200 mm long when complete (Taylor & Naish, 2007: table 1). As noted by Taylor & Naish (2007: 1554), ‘the length of the centrum, especially in so posterior a dorsal vertebra, argues against [a diplodocoid identity]: the posterior dorsal centra of diplodocoids typically have EI < 1.0, compared with 1.25 for R2095’—or 1.21 using the aEI of Chure et al. (2010: 384). However, rebbachisaurs may be unusual among diplodocoids in this respect—perhaps unsurprisingly, as they diverged early from the line leading to diplodocids, with their characteristically short dorsal centra, and likely retained something more similar to the ancestral neosauropod condition. Wilson & Allain (2015: 8) give the centrum measurements of MNHN MRS 1958 as posterior height 231 mm, posterior width 220 mm and length 220 mm. This yields an aEI of 0.98, meaning that the Xenoposeidon centrum is only 24% more elongate than that of Rebbachisaurus. This is a significant difference, but not an outlandish one. For comparison, the centrum of the basal rebbachisaurid H. boscarollii holotype ‘WN-V6’ is relatively elongate, with its posterior articular surface measuring 150 mm high and centrum length of ‘more than 200 mm’ (Dalla Vecchia, 1998: 122) yielding an EI of > 1.33. Also, the aEIs of the last four dorsal vertebrae of the Brachiosaurus altithorax holotype FMNH PR 25107 are 1.34, 1.27, 1.19 and 0.96 (calculated from the table of Riggs, 1904: 34): so aEIs of sauropod dorsals can vary, within two serial positions of the same individual, from values below that of MNHN MRS 1958 to above that of NHMUK PV R2095.

In conclusion, while the evidence regarding the serial position of NHMUK PV R2095 remains equivocal, it suggests a more posterior position than previous inferred—it can be fairly confidently described as ‘posterior’ within the broader ‘mid-to-posterior’ designation—but it is unlikely to be the very last dorsal.

Revised reconstruction

In light of the reassignment of Xenoposeidon to Rebbachisauridae, and the reinterpretation of its laminae, I present a new reconstruction of how the vertebra NHMUK PV R2095 might have looked when complete (Fig. 7). As in MNHN MRS 1958, the parapophysis and diapophysis are both elevated above the zygapophyses. The lateral CPRL and ACPL meet at a point where they project laterally about the same distance from the vertebra, as is apparent from the preserved portion of the vertebra; but the CPOL is assumed to pass through a sheet-like PCDL as in Rebbachisaurus, because it is clear from breakage in NHMUK PV R2095 that the PCDL extended further laterally from the body of the neural arch than the preserved portion indicates. The neural spine, composed as in Rebbachisaurus of pre- and post-spinal laminae together with the left and right SDLs, is shown fading out at the top, as there is no way to determine its height. The condyle that is the centrum’s anterior articular surface is reconstructed as only slightly convex, as in Rebbachisaurus. It is shown almost immediately anterior to the preserved portion of the centrum, because the camerae in the dorsal part of the anteriormost preserved portion reach their point of dorsalmost excavation a short distance behind the front part, indicating that the cortex at this point was curving down over the camerae to form the condyle.

Figure 7 NHMUK R2095, the holotype and only vertebra of Xenoposeidon proneneukos, in left lateral view, interpreted as a rebbachisaurid.

This interpretation is modelled primarily on MNHN MRS 1958, a posterior dorsal vertebra from the holotype specimen of Rebbachisaurus garasbae. The CPOL passes through a sheetlike PCDL, as in Rebbachisaurus; but the lateral CPRL forms a cross-shaped junction with the ACPL, each of these laminae equally interrupting the trajectory of the other. Scale bar = 200 mm.

It is instructive to compare this with the original reconstruction of the vertebrae (Taylor and Naish, 2007: fig. 5). The new reconstruction has a taller neural arch, a far more elevated parapophysis, a more posteriorly located diapophysis (no longer dorsal to the parapophysis) and a shallower condyle, as that of the original reconstruction was drawn with those of brachiosaurs in mind.

Systematic Palaeontology

Dinosauria Owen, 1842

Saurischia Seeley, 1888

Sauropodomorpha Huene, 1932

Sauropoda Marsh, 1878

Neosauropoda Bonaparte, 1986

Diplodocoidea Marsh, 1884

Rebbachisauridae Sereno et al., 1999

Xenoposeidon Taylor and Naish, 2007

Xenoposeidon proneneukos Taylor and Naish, 2007

Holotype. NHMUK PV R2095, the Natural History Museum, London. A mid-to-posterior dorsal vertebra consisting of partial centrum and neural arch.

Revised diagnosis: Differs from all other sauropods in the following characters: Neural arch covers dorsal surface of centrum, with its posterior margin continuous with that of the centrum.

Neural arch slopes anteriorly 30°–35° relative to the vertical.

Sharp oblique lamina above lateral fossa forms ventral border of a broad, flat area of featureless bone.

Very large, teardrop-shaped anterior fossa.

Arched laminae form vaulted boundary of anterior fossa, enclosed within the medial CPRLs.

Discussion

Age

As shown by Wilson & Allain (2015: table 1), the 19 then-recognised rebbachisaurids (of which 13 had been named) span the middle third of the Cretaceous. The earliest recognised taxon is Histriasaurus boscarollii from the upper Hauterivian or lower Barremian limestones of southwest Istria, Croatia (Dalla Vecchia, 1998). Seven taxa, of which five are named, survived at least to the Cenomanian (earliest Late Cretaceous), of which two—Katepensaurus goicoecheai and Limaysaurus tessonei—may be from the Turonian (Ibiricu et al., 2013, 2015; Salgado et al., 2004; Garrido, 2010).

As discussed by Taylor & Naish (2007: 1547–1548), the precise location and horizon where NHMUK PV R2095 was excavated were not recorded in the specimen’s original brief description, which only said ‘the Wealden of Hastings’ (Lydekker, 1893: 276). However, records of the collection of Philip James Rufford, who collected the specimen, indicate that the most likely location is Ecclesbourne Glen, a mile or two east of Hastings, East Sussex (see discussion in Taylor & Naish, 2007: 1548). The units exposed at Ecclesbourne Glen are part of the Ashdown Formation (formerly the Ashdown Beds Formation), which straddles the Berriasian/Valanginian boundary; but the part of the formation at that location is from the earlier Berriasian age. If this assessment is correct, then Xenoposeidon is from the very earliest Cretaceous, giving it an age of around 140 million years—about 10 million years earlier than Histriasaurus.

As the oldest known member of Rebbachisauridae, Xenoposeidon is an important taxon for understanding the evolution and palaeobiogeography of the group. The apparent paucity of rebbachisaurs in Gondwana during the early Early Cretaceous may merely be an artifact of an incomplete fossil record. On the other hand, the distribution of rebbachisaurids across South America and Africa in the mid-Cretaceous could represent the flowering of a lineage extending back into the earliest Cretaceous or Late Jurassic in Europe, suggesting a migration from the northern supercontinent of Laurasia to the southern supercontinent of Gondwana.

Within Rebbachisauridae, the early age of Xenoposeidon is consonant with a basal position. However, further material will be required before numerical phylogenetic work can firmly establish its position within the group.

Wealden rebbachisaurs

Although Xenoposeidon is the first named rebbachisaurid from the Wealden Supergroup, other material from this unit has been referred to Rebbachisauridae. Naish & Martill (2001: plate 36, opposite page 236) illustrated some isolated sauropod teeth IWCMS.2001.201–203, and these were referred to Rebbachisauridae by Sereno & Wilson (2005: 174). Mannion (2009) described a partial rebbachisaurid scapula MIWG 6544. Finally, Mannion, Upchurch & Hutt (2011) described a proximal caudal neural arch MIWG 5384, which they also interpreted as rebbachisaurid. All of these specimens are from the Barremian Wessex Formation of the Isle of Wight, so they could all belong to the same species or genus. However, since the likely Berriasian age of NHMUK PV R2095 makes it 10–15 million year older than these specimens, it is unlikely that they belong to Xenoposeidon, but to some other as yet-unnamed rebbachisaurid. Thus, is likely that the Wealden Supergroup contains at least two rebbachisaurid sauropods.

3D models of complex bones

Electronic 3D models were invaluable in determining Xenoposeidon’s true affinities. Most obviously, the model of the Xenoposeidon vertebra itself, created by Heinrich Mallison (Palaeo3D), has functioned as an invaluable proxy for the fossil itself when I am unable to visit the NHMUK, and I have consulted it many times in writing this paper. I would also have been unable to determine to my own satisfaction whether the Katepensaurus dorsals feature intersecting laminae like those of Rebbachisaurus without the models provided by Lucio M. Ibiricu. Although no true model is available for the Rebbachisaurus dorsal itself or for the dorsal vertebrae of Nigersaurus, rotating videos were crucial in enabling me to understand their morphology. When interpreting specimens for which no such models exist, such as Russell’s (1996) referred Rebbachisaurus specimen NMC 50844, the conclusions reached using only 2D representations—whether photographs or drawings—are much less well founded.

Techniques such as photogrammetry (see Falkingham, 2012; Mallison & Wings, 2014) are reducing the barriers to the creation of high-quality 3D models in full colour. Doing so is now inexpensive in both time and money. In light of our discipline’s goal of making palaeontology more accessible and reproducible, then, it should become increasingly routine in the 21st Century to provide 3D models as a standard part of the description of complex bones such as sauropod vertebrae.

Supplemental Information

Supplemental Information 1 Supplemental File.

Click here for additional data file.

I thank Sandra D. Chapman (Natural History Museum, London) for access to the Xenoposeidon specimen, and Heinrich Mallison (Palaeo3D) who went far beyond the call of duty in building the 3D model of NHMUK PV R2095, supplying that of the Giraffatitan vertebra MB.R.3822, and talking me through aspects of photogrammetry. Lucio M. Ibiricu kindly provided access to unpublished 3D models of an anterior dorsal neural arch and a more posterior dorsal vertebra of Katepensaurus. I am also grateful to Jeff Wilson (University of Michigan) and Ronan Allain (Muséum National d’Histoire Naturelle, Paris) for sharing high-resolution photographs of the French Rebbachisaurus vertebra, and to Mathew J. Wedel (Western University of Health Sciences) and Darren Naish (University of Southampton) for helpful discussion. Phil Mannion (Imperial College London), Daniela Schwarz (Museum für Naturkunde Berlin) and Lucio M. Ibiricu provided constructive, detailed reviews that have helped to strengthen the arguments made herein; I also thank an additional anonymous reviewer.

As noted in Taylor (2015), this project began when I recognised the true identity of the curious laminae on the Xenoposeidon vertebra while viewing a rotating video of the R. garasbae holotype dorsal vertebra MNHN MRS 1958 on the University of Michigan Museum of Paleontology’s UMORF web-site (University of Michigan Online Repository of Fossils) at https://umorf.ummp.lsa.umich.edu/wp/gallery/vertebrate-animations/. This video was based on a 3D reconstruction created from CT scans performed at the AST-RX (Accèes Scientifique à la Tomographie à Rayons X) of the MNHN by F. Goussard.

Anatomical Abbreviations

aEI average elongation index sensu Chure et al. (2010: 384): length of a centrum divided by the average of the height and width of the posterior articular surface

ACPL anterior centroparapophyseal lamina

CPOL centropostzygapophyseal lamina

CPRF centroprezygapophyseal fossa

CPRL centroprezygapophyseal lamina

EI elongation index sensu Wedel, Cifelli & Kent Sanders (2000): length of a centrum divided by the height of the posterior articular surface

PCDL posterior centrodiapophyseal lamina

PCPL posterior centroparapophyseal lamina

POSL postspinal lamina

Postzyg postzygapophysis

PPDL paradiapophyseal lamina

Prezyg prezygapophysis

PRPL prezygaparapophyseal lamina

PRSL prespinal lamina

SDL spinodiapophyseal lamina.

Institutional Abbreviations

GSGM Gansu Geological Museum, Gansu Province, China

IWCMS Isle of Wight County Museum Service at Dinosaur Isle, Sandown, Isle of Wight, England

HMN Humboldt Museum für Naturkunde, Berlin, Germany

MBR Museum für Naturkunde Berlin, Berlin, Germany (fossil reptile collection)

MIWG Museum of Isle of Wight Geology (now Dinosaur Isle Visitor Centre), Sandown, Isle of Wight, England

MNHN Muséum National d’Histoire Naturelle, Paris, France

NHMUK the Natural History Museum, London, England

NMC Canadian Museum of Nature (previously National Museum of Canada), Ottawa, Ontario, Canada

‘WN’ ‘without number,’ an informal designation for specimens awaiting accession.

Additional Information and Declarations

Competing Interests

Author Contributions

Data Availability

The authors declare that they have no competing interests.

Michael P. Taylor conceived and designed the experiments, performed the experiments, analysed the data, contributed reagents/materials/analysis tools, prepared figures and/or tables, authored or reviewed drafts of the paper, approved the final draft.

The following information was supplied regarding data availability:

Taylor, Mike (2017): 3D surface scan of Xenoposeidon proneneukos NHMUK R2095. figshare. https://doi.org/10.6084/m9.figshare.5605612.v2.

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
