# Peer review of "Xenoposeidon is the earliest known rebbachisaurid sauropod dinosaur"

_PeerJ, doi:10.7717/peerj.5212_

## Round 0.1 · original submission · Major Revisions

Dear author,

As the three reviewers have not made a uniform recommendation, I have accepted the decision of 'major revision' from two of the reviewers.

Please read carefully through the comments of the reviewers. I would like to highlight that you raise the uncertainty in the age of the holotype more in the manuscript. Furthermore, based on the comments of reviewers one and two, is it possible the specimen is a nomen dubium, and best referred to as Rebbachisauridae indet.? Something to consider given their comments on the preservation and questions on the possible autapomorphies.

Once again, thank you for submitting your manuscript to PeerJ and I look forward to receiving your revision.

Reviewer 1 ·

Basic reporting

The authors made a re-interpretation of a fragmentary sauropod vertebra, previously erected as Xenoposeidon by the author of this manuscript as the leader author.
This taxon was subsequently considered as nomen dubium.
The english is clear, but the manuscript is wrote mainly as an opinion note. I personally think that most of it should be re-writed. I have the feeling that in some parts there is some circularity in the ideas exposed, this is explained below. For example a really important aim of the manuscript, which is defend the validity of the species, is presented in the introduction. This is, probably, one of the most important aims that the author should adress, and is not correct to mention this point in the introduction becouase gives the false impression that no detailed explanations are needed, that sounds some pedant. I personally believe that most of the diagnostic characters falls to be clear and that some of them are product of preservation. This was mentioned by other authors and a proper discussion is deserves.
I made some comments on the manuscript.

Experimental design

In this paper the element, holotype of Xenoposeidon, is re-interpreted and the validity of the taxon is reinforced. In the base of the new laminae interpretation, based on Rebbachisaurus, the authors indicates the rebbachisaurid affinities of Xenoposeidon (I have the impression that some circularity is present in that).
In my opinion the manuscipt, deserves, at least, some extra work.
The M pattern observed in the fragmentary vertebra is also present in some other sauropods (see for example Neuquensaurus), which laminae are different to those of Rebbachisaurus. I see the similarities between Xenoposeidon and Rebbachisaurus, but I also feel that the material is to fragmentary to clearly refer this taxon to this clade. At least some phylogenetic analysis could be made, but again the fragmentary nature of the specimen precludes to unambuguosly reinterpret the laminae.

Validity of the findings

As explained above I am not convinced on the results. Briefly, the weakness points are:

1-The vertebra is too fragmentary, despite the validity of the taxon is defended. Its laminae patter is not as obvious as the author statted.

2-The diagnosis is not very well explained and some characters looks like prodcut of breakage (dorsal neural arch covering the dorsal surface of the centrum, which should be posterior as the anterior one is much damaged) or the possibility of incorrect orientation of the element (the inclination of the anterior edge).

3-The phylogenetic interpretation (as a rebbachisaurid) is made in the base of the laminae interpretation made with a rebbachisaurid.... .

4-The main goal of the manuscript is based on the phylogenetic interpretation (see above) and the age of the specimen. The age informed is also some dubious as not exact position is know, and although the age proposed sounds like the most probable, is not possible to strongly confirm this.

Annotated reviews are not available for download in order to protect the identity of reviewers who chose to remain anonymous.

·

Basic reporting

I have only a positive comment here: the inclusion of 3D data of vertebral specimens in this paper is very helpful and should be supported. It makes the paper more transparent, as characters can be easier checked.

Experimental design

no comment

Validity of the findings

In this manuscript, a dorsal vertebra of a sauropod dinosaur, described in 2007 by Taylor & Naish, is re-interpreted and re-described. The work is based on comparison with dorsal vertebrae of rebbachisaurid sauropods (in particular Rebbachisaurus garasbae), for showing a unique "M" shared arrangement of laminae associated with the parapophysis and the vertebral centrum. The author of the manuscript challenges here some of his earlier interpretations of lamina arrangement (and determination) and position of the parapophysis and thus revises his own work in a positive way.
Given the doubts on the validity of the described vertebra NHMUK R2095 (e.g., D’Emic 2012) and the limited anatomical information available from the fragmentary specimen, this manuscript is very helpful as it resolves the interpretation of the lamina system at the lateral vertebral centrum; helps to understand its morphology better; and gives new insights into the systematic relationships of Xenoposeidon.

The paper discusses four main issues that I want to comment on:
1. I do strongly agree with the new interpretation of the vertebral laminae as discussed here and the interpretation of the specimen to belong to a rebbachisaurid sauropod. This alone would be worth publishing and as the attached figures underline, is very obvious. I would suggest that the description of the lamina system in Rebbachisaurus garasbae could be a bit more detailed, for example, how does the lamina system change throughout the dorsal vertebral column of this taxon, or is there always an “M” shape present?

2. I disagree on the interpretation of the position of the parapophysis to be elevated above the level of the prezygapophysis. In the described specimen, both elements are not preserved (indeed, most of the neural arch is missing) and this means that the position of zygapophyses and parapophyses cannot be reconstructed. I consider it not appropriate to base the interpretation of the specimen's anatomy on an interpretative reconstruction of in dependence on a complete rebbachisaurid vertebra. Even if the arrangement of laminae is similar in both specimens, this does not mean that the position of the parapophysis is, too. Although it cannot be excluded that the parapophysis was positioned above the level of the prezygapophysis, there is no possibility to positively demonstrate that it was the case, and this condition should simply not be used as a character. Additionally, as Fig. 4 clearly shows, the 2D lateral view seems to indicate an elevated parapophysis, but this can also be just in reference to the perspective of the drawing (by showing the laminae ventrally to the transverse process). Only an anterior or posterior view can show the pristine elevation of the parapophysis. I think that the parts of the discussion basing on the elevation of the parapophysis in the described vertebra should be omitted.

3. With respect to the missing zygapophyses and parapophyses, I suggest to revise the interpretation of the anatomical position of the specimen in the dorsal vertebral column. As the author mentions in the paper, the described vertebra is longer than a rebbachisaurid vertebra in the posterior dorsal region. The determination as a posterior dorsal vertebra is based on the placement of the parapophyses, which are not preserved; however, the elongation of the vertebral centrum (which is preserved) allows an alternative interpretation as mid-dorsal vertebra. I suggest to at least discuss/consider the alternative possibility that this vertebra is a mid-dorsal one instead of a posterior dorsal vertebra. Would this have an effect on the interpretation of the laminae?

4. I find it difficult to understand in which respect the revised diagnosis differs from the original one (both should be compared here, given that not all readers will know the paper of Taylor & Naish 2007) except to the new interpretation of the laminae. The five characters listed for the revised diagnosis of this paper are more or less undiscussed and in this form do not provide definitive evidence for the validity of the taxon Xenoposeidon per se. If the validity of the taxon is to be demonstrated here, there should be a slightly more detailed description and comparison of the listed autapomorphies to at least some other sauropods. This does not say that I doubt the validity of Xenoposeidon, but I consider it helpful to get a somewhat sharpened discussion here. Parts of the introduction, discussing the validity of the taxon with respect to the paper of D’Emic 2012, should be moved into this part of the discussion. It would be great also to have these characters outlined in one of the figures.

·

Basic reporting

The paper is well-written throughout, although there are a few instances where other references should be cited (see detailed annotations on the MS). My only real criticism of the Basic reporting is that there is little in the way of background/context. In some ways the current Introduction is welcome in that it gets straight to the point, but it feels unusual in that there is so little of the usual kind of background information, e.g. that sauropods were dominant herbivores in the Mesozoic, etc. It would be worth considering adding some kind of introduction that both sets the general scene, but that also places Xenoposeidon in some sort of context: i.e. why is it remotely important/interesting to know what Xenoposeidon is? The earliest Cretaceous is a bit of a blank in most regions for sauropods (and for most groups in general), and so it's at least important for helping us understand more about that time interval. The long rebbachisaurid ghost lineage is also another important and relevant issue.

Also, perhaps consider adding a short Conclusions section.

Experimental design

No issues here, other than by not including some kind of Background/Context (see above), the MS does not really state why this work is relevant and meaningful, or how this fills a research gap. I believe that it is all of these, but currently the MS does not do a good job of conveying that significance.

Validity of the findings

Although I agree with the main conclusion of the MS (i.e. that Xenoposeidon is likely a rebbachisaurid, and thus the earliest known member of this clade), I disagree or have reservations about a number of the other reported findings. All of these are documented in the annotated version of the MS, but I've summarised the main ones below:

(1) I'm unconvinced by all five proposed autapomorphies for Xenoposeidon:

Autapomorphy #1 - It looks like the neural arch continues to the posterior margin of the centrum in Rebbachisaurus too.

Autapomorphy #2 - The strong anterior slant of the neural arch appears to be dependent on how you've chosen to orientate the vertebra, but there doesn't appear to be any need to orientate it in this way (expanded upon in the annotated MS).

Autapomorphy #3 - How does the "broad, flat, featureless bone on the lateral face of neural arch" differ from the condition in Rebbachisaurus or Nigersaurus?

Autapomorphy #4 - The teardrop-shaped centroprezygapophyseal fossa does not appear to be all that different to the one in Rebbachisaurus.

Autapomorphy #5 - The arched laminae that form the boundary of the centroprezygapophyseal fossa (CPRF) are the medial CPRLs in Rebbachisaurus. By definition, the CPRF has to be bounded by centroprezygapophyseal laminae (see Wilson et al. 2011). If we follow Wilson & Allain (2015) for Rebbachisaurus, then their medial CPRLs form the margins of the CPRF, meeting at the CPRF apex and contacting the prezygapophyses (required for them to be interpreted as CPRLs in the first place). In most other sauropods, they couldn't be called CPRLs in this instance, because the prezygapophyses would be interrupted by a TPRL, but the zygapophyses meet at their midline in rebbachisaurids. As such, why can't these arched laminae in Xenoposeidon be medial CPRLs, meeting the (unpreserved) prezygapophyses at their midline? If you maintained the orientation of the unpreserved portion of what you are interpreting as the lateral CPRL such that it follows that of the preserved portion (rather than deflecting it dorsally in your reconstruction), it looks like all of these laminae end up at roughly the same height on the vertebra. A remaining issue though would be that this would mean that you have three CPRLs but, if the lateral CPRL is correctly identified, then the medial CPRL could be interpreted as being excavated/bifurcated, resulting in a medial centroprezygapophyseal lamina fossa (mCPRL-F), following the nomenclature of Wilson et al. (2011). This bifid/excavated mCPRL might then be considered an autapomorphy...

(2) A couple of remaining putative differences between Xenoposeidon and other rebbachisaurids can probably be readily resolved. The elongate centrum doesn't appear to be much different from a posterior dorsal vertebra of Nigersaurus (http://digimorph.org/specimens/Nigersaurus_taqueti/dorsal_vertebra/). I also don't think that Xenoposeidon necessarily lacks a strongly arched ventral margin, once you take into account that the ventral margin of the posterior cotyle is incomplete and the anterior end of the centrum is missing. There's every possibility that it might have looked like a slightly elongate version of the Rebbachisaurus vertebra illustrated in Fig. 4D of Wilson and Allain (2015).

(3) The lateral pneumatic foramen of Xenoposeidon is set within a distinct fossa, similar to that of many somphospondylans, and is very different to the condition in Rebbachisaurus - I don't think that they can or should be described as comparable, and would be better kept as a notable difference (possibly as a local autapomorphy of Xenoposeidon).

(4) It's stated that the internal tissue structure of Xenoposeidon differs from the camellate condition of most rebbachisaurids (lines 318-320); however, this is incorrect: there is no documented evidence for camellae in any rebbachisaurid dorsal vertebra: they all seem to have a camerate internal tissue structure.

Additional comments

This is a welcome reinterpretation of Xenoposeidon, and I am in agreement that a rebbachisaurid identification appears likely. In addition to the above comments, I have made a number of comments/suggestions on an annotated PDF of the MS, as well as spotting a few typos.

I look forward to seeing the final version of this paper.

Best wishes,

Phil Mannion

---

## Round 0.2 · Minor Revisions

Dear author,

I have accepted the decision of 'minor revisions' from the reviewers. Please note I had to get a new second reviewer (reviewer 4), as I was unable to get reviewers 2 and 3 from the previous round to re-review the manuscript.

Please note that the specimen number should read as NHMUK PV R 2095 (PV stands for Palaeontology Vertebrate, which the NHMUK added to their specimen numbers). (see: http://data.nhm.ac.uk/dataset/56e711e6-c847-4f99-915a-6894bb5c5dea/resource/05ff2255-c38a-40c9-b657-4ccb55ab2feb/record/467030).

Once again, thank you for submitting for manuscript to PeerJ, and I look forward to receiving your revised version.

·

Basic reporting

I'm much happier with this aspect of the revised MS, and have no further comments regarding Basic reporting.

Experimental design

I'm also happy with this

Validity of the findings

I'm still unconvinced by the proposed anterior slant of the vertebra and don't think that there's any evidence for orientating it in this way. I went into the NHM to re-look at this. No aspect of the posterior articular surface of the centrum leads me to orient the vertebra in the same way of shown in your figures. In addition, as currently orientated, the floor of the neural canal is strongly tilted - it seems more conservative to assume that this is horizontal. Similarly, by following that orientation, this would then make the long-axis of the lateral pneumatic opening closer to horizontal. By orientating the vertebra this way, the anterior margin is sub-vertical, with a very gentle anterior deflection (i.e. fairly normal for a sauropod), and the M-lamina is much closer in orientation to that of Rebbachisaurus.

I also remain unconvinced by the interpretation of what's going on with the laminae that frame the CPRF, as well as the reasoning that these can't be bifid medial CPRLs based on the inferred position of the prezygapophyses. Do the prezygapophyses really have to be placed so high? If you continue the chords for the mCPRL and lCPRL in the same orientation as the preserved portions of the laminae, they meet much lower down than currently inferred in Fig. 5. Also, rebbachisaurid zygapophyses are strongly tilted - see your reproduction of a Rebbachisaurus dorsal in anterior view - so where the two prezygapophyses meet (it's unlikely there would have been a TPRL) is much lower than where the lCPRL would contact the prezygapophysis. Also, as previously mentioned, if the CPRF isn't bound by CPRLs, then it technically isn't a CPRF.

My other comments are very minor and are included in an attached annotated version of the MS.

Additional comments

Aside from those contested features, I'm essentially happy with the revised MS, and remain in agreement that Xenoposeidon is probably a rebbachisaurid.

Best wishes,

Phil Mannion

·

Basic reporting

I am not a native English speaker; however, the English appear to be appropriate for a scientific manuscript. The manuscript includes the majority of the articles related to the group; therefore, the literature is well referenced (with few exceptions, see annotated pdf). Likewise, the rebbachisaurid taxa are all included. The figures described correctly the aim of the manuscript, particularly after the inclusion of the figures suggested by reviewers. In agreement with the author, the 3D model is an important tool, which helps to better understand morphology of the dorsal vertebra (see annotated pdf).

Experimental design

The main aim of the manuscript is to support the inclusion of R2095 within Rebbachisauridae. It is well-supported (throughout the text and figures), however, because the material is fragmentary in many aspects is “controversial” (see annotated pdf).

Validity of the findings

The manuscript is well stated and it is directly linked to the main question about the assignation of the material as a rebbachisaurid sauropod. Additionally, the different points of view regarding this material is mentioned in the manuscript (for example those of Upchurch et al., 2011; D’Emick 2012 and Mannion et al., 2013) leaving this controversy open to the reader (an interesting arguable topic). I think that the author may have to go deeper in some aspects. He supported the inclusion of BMNH R2095 within Rebbachisauridae and mentioned in the title (“the earliest known rebbachisaurid”) and in the introduction (earliest Cretaceous sauropods are important for our understanding of the evolution). Nevertheless, despite some phrases throughout the text, there are not a detailed or more clear explanation about the importance that its assignation has, considering that BMNH R2095 is now the oldest representative of Rebbachisauridae (see pdf). I think that could be interesting to analyze this topic deeply, which may enrich the manuscript and the impact on the group (i.e., within Diplodocoidea in general and Rebbachisauridae in particular). Comments within the pdf (particularly, in Discussion section) may help on this topic.

Additional comments

The fossil record of Rebbachisauridae is scarce compared to other groups of sauropods (inclusive within Diplodocoidea). Therefore, every new record of this clade adds important information increasing our knowledge on the group. The fragmentary nature of BMNH R2095 difficult its assignation (mentioned in the ms in 2007/2011/2012/2013 papers). However, the manuscript added important information to this topic, and leads to continue the discussion about it. The putative assignation of BMNH R2095 within Rebbachisauridae is important because it is (now) the oldest representative of the group. Therefore, it has a direct relation with the evolutionary dynamic of Rebbachisauridae. In sum, the manuscript, after the inclusion of several points raised by the reviewers (particularly the new figure) improved substantially. However, please, see annotations on the pdf.

---

## Round 0.3 · accepted · Accept

Dear author,

Many thanks for your revised manuscript. I have accepted the reviewers decision of 'accept'.

Once again, thank you for submitting your manuscript to PeerJ and I hope you will use us again as your publication venue.

If we need to clarify any details required to move the manuscript forward, then our production staff will get in touch with you. Otherwise, a proof will be forthcoming shortly for your review.

Congratulations and thank you for your submission.

# ·

Basic reporting

No comment

Experimental design

No comment

Validity of the findings

No comment

Additional comments

I'm happy to agree to disagree on what are relatively minor aspects of anatomy that do not impact upon the proposed placement of Xenoposeidon as a rebbachisaurid.

Best wishes,

Phil